# Granularity Matters: Pathological Graph-driven Cross-modal Alignment for Brain CT Report Generation

**Yanzhao Shi[1], Junzhong Ji[1], Xiaodan Zhang[1,\*], Liangqiong Qu[2,\*], Ying Liu[3]**

[1]Faculty of Information Technology, Beijing University of Technology, Beijing, China
[2]Department of Statistics and Actuarial Science, University of Hong Kong, Hong Kong, China
[3]Department of Radiology, Peking University Third Hospital, Beijing, China
zhangxiaodan@bjut.edu.cn, liangqqu@hku.hk

## Abstract

The automatic Brain CT reports generation can improve the efficiency and accuracy of diagnosing cranial diseases. However, current methods are limited by 1) coarse-grained supervision: the training data in image-text format lacks detailed supervision for recognizing subtle abnormalities, and 2) coupled cross-modal alignment: visual-textual alignment may be inevitably coupled in a coarse-grained manner, resulting in tangled feature representation for report generation. In this paper, we propose a novel Pathological Graph-driven Cross-modal Alignment (PGCA) model for accurate and robust Brain CT report generation. Our approach effectively decouples the cross-modal alignment by constructing a Pathological Graph to learn fine-grained visual cues and align them with textual words. This graph comprises heterogeneous nodes representing essential pathological attributes (*i.e.*, tissue and lesion) connected by intra- and inter-attribute edges with prior domain knowledge. Through carefully designed graph embedding and updating modules, our model refines the visual features of subtle tissues and lesions and aligns them with textual words using contrastive learning. Extensive experimental results confirm the viability of our method. We believe that our PGCA model holds the potential to greatly enhance the automatic generation of Brain CT reports and ultimately contribute to improved cranial disease diagnosis.

## 1 Introduction

Writing diagnostic reports for Brain CT imaging with multiple scans is widely applied in medicine, to summarize the findings of cranial diseases. Nonetheless, this traditional clinical practice could be time-consuming and error-prone for radiologists due to some subjective factors (*e.g.* fatigue and distraction) (Brady et al., 2012). A computer-aided

---

*Corresponding Authors

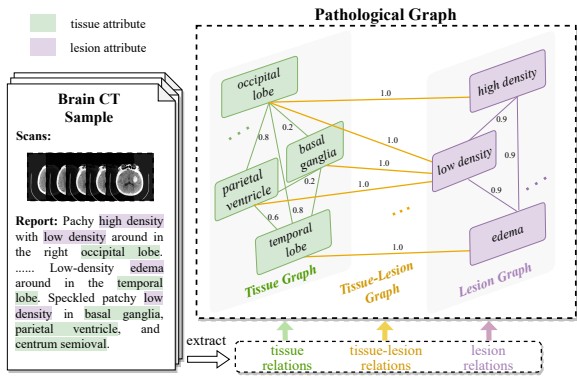

Figure 1: An example of the pathological graph. Pathological elements (nodes) are summarized from reports as tissue attribute and lesion attribute, which are connected as the tissue graph, lesion graph, and tissue-lesion graph with prior relations (edges). These three sub-graphs compose the fine-grained pathological graph.

reporting system aims to automatically generate accurate reports, which has the potential to lighten the workload of physicians and economize insufficient clinical resources in populated areas.

With the advent of deep neural networks in image captioning (Vinyals et al., 2015; Anderson et al., 2018), medical report generation (MRG) methods are increasingly ramping up (Jing et al., 2018; Wang et al., 2018; Li et al., 2019; Yang et al., 2021; Yan et al., 2021; Chen et al., 2021; Song et al., 2022; Qin and Song, 2022; Yang et al., 2023; Li et al., 2023). Different from image captioning, MRG task focuses on subtle yet crucial medical terminologies, with the report length typically 4-6 times longer than those of nature object captions. This prompts MRG models to refine the dedicated consistency of salient pathological features between visual and textual modalities. To achieve this, recent studies have employed various cross-modal alignment mechanisms (Liu et al., 2021; Wang et al., 2022; Li et al., 2023), leveraging specific knowledge to effectively improve report generation.

Despite the promising achievements, previous methods are still thwarted by the following two concerns. 1) Coarse-grained supervision: In clinical practice, brain findings are often characterized by some subtle yet vital pathological elements that belong to different attributes (i.e. tissues and lesions)(Griffin et al., 2002), see Figure 1. However, medical images and long reports are always treated as coarse-grained supervisory signals in the mainstream methods. Although few recent studies have explored the learning of fine-grained signals (Liu et al., 2021; Wang et al., 2022) in chest X-ray samples, it is unsuitable for Brain CT which encompasses more sophisticated pathology patterns. Thus, how to leverage the intrinsic fine-grained cranial knowledge to recognize subtle abnormalities remains an open question. 2) Coupled cross-modal alignment: Current cross-modal alignment methods in MRG inevitably tend to couple visual-textual representation during coarse-grained alignment, which is manifested as highly-similar attention maps (Yang et al., 2021) or tangled semantic representations in instance-level contrastive learning (Yan et al., 2021), resulting in inadequate feature learning.

In this work, we propose a novel Pathological Graph-driven Cross-modal Alignment model, named PGCA, which introduces a detailed Pathological Graph (PG) to seamlessly capture domain knowledge in samples, and explicitly extract fine-grained semantic correspondence between visual regions and diagnostic texts for accurate Brain CT report generation. Specifically, the PG contains heterogeneous nodes of tissue and lesion attributes, which are connected by edges with prior knowledge. As shown in Figure 1, we decompose PG into two fixed Tissue Graph and Lesion Graph with intra-attribute knowledge (relations in each attribute), and one dynamic Tissue-Lesion Graph with inter-attribute knowledge (correspondences between two attributes for each case). Then, we respectively incorporate Tissue and Lesion Graphs into an attribute-oriented graph convolution network for dedicated node features learning, which is jointly updated via an Intra-Attribute Classification (IAC) loss module. While the Tissue-Lesion graph is updated via an Inter-Correlation Alignment (ICA) loss module, which facilitates learning the complex tissue-lesion connections, in turn, benefits node feature representation. Finally, a cross-modal Contrastive Learning (CL) module is pro-

posed to reconcile the learned visual features with their corresponding word embeddings in the report, to further boost the visual encoder and textual decoder in report generation.

In sum, our main contributions contain:

1. We propose a novel framework to seamlessly capture detailed domain knowledge from Pathological Graph, and explicitly align fine-grained visual and textual features of pathology, which improves the encoder and decoder of Brain CT report generation by sharing feature learning layers.

2. We, for the first time, introduce the idea of decoupling cranial tissues and lesions via Pathological Graph into the medical report generation area, which is capable of handling fine-grained alignment between long-text report and multiple scans.

3. We comprehensively validate our model on the BCT-CHR dataset. The experimental results indicate that our method surpasses previous arts in medical report generation.

## 2 Related Work

### 2.1 Medical Report Generation With Knowledge Graph

To mimic the expert knowledge of radiologists, the using of Knowledge Graph (KG) can endow the MRG model with better capabilities of abnormal recognition and has gained increasing research interest, which can be summarized into three types. The first type focuses on stressing clinical terminologies (e.g. abnormalities and diseases). Li et al. (2019) extracts a set of chest medical terminologies from the MIMIC-CXR dataset, which are regarded as graph nodes. The edges linking nodes are assigned with attention weights to depict latent relations, which may be affected by error-updated cases. The second type is to build a universal graph that contains lesion nodes with stable edges acquired by prior knowledge (Zhang et al., 2020; Liu et al., 2021). Such relationships are stored in the adjacency matrix and learned by graph convolution networks. As an extension of this graph, Li et al. (2023) proposes to dynamically add characterized nodes for each sample, but still limits to represent another detailed attribute, i.e. pathological tissues. The last type utilizes NLP-rule based methods to extract detailed triplets from the training corpus

and build a clinical graph (Li et al., 2022). The sample-related triplets are restored to learn specific knowledge, though the triplets may be wrong extracted.

Our pathological graph seamlessly combines the advantages of the last two types. We first separately construct two fixed tissue and lesion graphs as the way in the second type for dedicated pathology representing, and then extract structured triplets for each case to depict fine-grained relations between paired attributes.

## 2.2 Cross-modal Feature Alignment

Learning medical semantics across visual and textual modalities is essential for MRG model to generate logical and accurate reports. Thus, the cross-modal feature alignment has attracted growing interest in recent studies, which can be roughly divided into three stages. In the early stage, various attention mechanisms (Jing et al., 2018; Wang et al., 2018; Xue et al., 2018; Yang et al., 2021) are adopted to extract abnormal visual features and guide the generation of diagnostic texts. However, the attention map updated by naive cross-entropy loss could not sufficiently represent the complex cross-modal patterns. For the second stage, Chen et al. (2021) introduces the memory vector to restore multi-modal relations, which is further extended with reinforcement learning (Qin and Song, 2022) and class-related prototypes (Wang et al., 2022). Since the visual and textual features across different samples are highly similar, it is still challenging to capture the essence of abnormal clues. Parallel to the second stage, the incorporation of contrastive learning can effectively distinguish similar features, which paves the next phase. Yan et al. (2021); Yang et al. (2023) utilize instance-level contrast (i.e. image-report pairs from same cases are positives, otherwise are negatives) in MRG models to enhance the consistency of multi-modal features. Li et al. (2023) further improves the visual representation with graph features and benefits the report generation. Our PGCA follows the third stage. Different from previous methods, we refine complex visual features into dedicated node features, which are contrasted with related medical word features for fine-grained feature alignment.

## 3 Methodology

As shown in Figure 2, our framework contains two parallel branches, namely Brain CT report gener-

ation and Pathological Graph-driven Cross-modal Alignment (PGCA), which interact by the shared visual and textual embedding layers.

### 3.1 Brain CT Report Generation

Given the input scans $S = \{s_1, ..., s_N\}$, where $N$ denotes the number of scans in each sample, the target of this branch is to generate a diagnostic report $Y = \{y_1, ..., y_M\}$ with $M$ words. The report generation model follows the traditional encoder-decoder pipeline. Firstly, we apply the ResNet101 (He et al., 2016) to extract visual features of $S$ that contain global features $F = \{f_1, ..., f_N\} \in \mathbb{R}^{N \times d}$ ($N = 24$, $d = 2048$) and grid features $G = \{g_1, ..., g_N\} \in \mathbb{R}^{N \times H \times d}$ ($H = 196$). Then, $F$ and $G$ are embedded by a visual encoder, resulting in global and spatial visual feature embeddings $V_f$ and $V_g$, respectively. Finally, visual features $V = \{V_f, V_g\}$ is used to generate long reports in the decoder, which contains a textual embedding layer and a language model with keywords-driven interactive recurrent network (Yang et al., 2021). We train the parameters $\theta$ by minimizing the cross-entropy loss, which can be expressed as:

$$\mathcal{L}_g = -\sum_m^M \log p_\theta(x_m | V, x_{1:m-1}), \qquad (1)$$

where $p(x_m | V, x_{1:m-1})$ denotes the predicted probability for the *m-th* word based on visual features $V$ and previous word embeddings $x_1, x_2, ..., x_{m-1}$.

### 3.2 Pathological Graph-driven Cross-modal Alignment

To boost the report generation, we propose to learn fine-grained visual-textual representations to improve the encoder and decoder by co-training with PGCA, which contains graph construction, graph embedding and updating, and cross-modal contrastive learning.

#### 3.2.1 Pathological Graph Construction

The knowledge graph is widely adopted to represent the relationship of medical entities. Different from previous Chest report generation methods (Zhang et al., 2020; Liu et al., 2021) with only 1-2 images and fewer pathological entities for diagnosis, generating Brain CT reports from multiple scans meets more challenges. Regularly, elements to be reported from scans contain some key

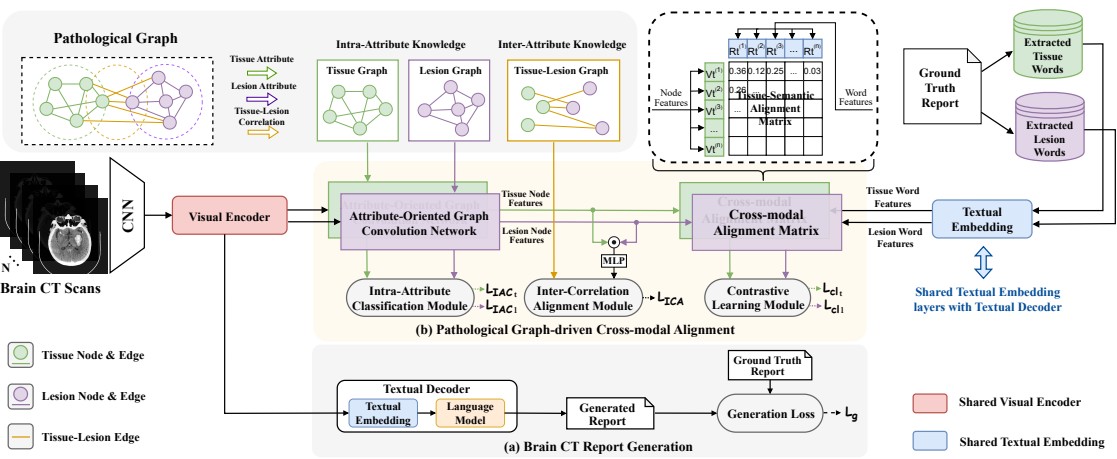

Figure 2: An overview of the proposed model, which comprises (a) Brain CT report generation and (b) Pathological Graph-driven Cross-modal Alignment. Communicated by the shared visual and textual embedding layers, both (a) and (b) are simultaneously used in the training stage while only (a) is used for testing.

brain tissues and their associated lesions, which are depicted as fundamental brain findings (Griffin et al., 2002). Motivated by this, we propose to organize a Pathological Graph (PG) to cover clinically important elements in Brain CT samples. We first obtain a set of critical tissues and lesions, denoted as $T = \{t_1, ..., t_{N_t}\}(N_t = 19)$ and $L = \{l_1, ..., l_{N_l}\}(N_l = 13)$ respectively, where $N_t$ and $N_l$ are the numbers of selected tissues and lesions , based on the knowledge of experts and the word frequency in training corpus. These elements serve as the nodes in our PG. Given the heterogeneous nature of the nodes (i.e., tissue and lesion attributes), we proceed to link them using edges derived from various prior knowledge, such as relations in tissues, relations in lesions, and relations between tissues and lesions, as shown in Figure 1. This process effectively partitions the PG into three distinct sub-components: the Tissue Graph, Lesion Graph, and Tissue-Lesion Graph. Next, we mainly introduce the building of these three components.

1) *Tissue Graph*: Clinically, physicians often divide Brain CT scans into 8 layers to facilitate the diagnosis of different brain tissues. Following this diagnostic pattern, we enhance the correlation of tissues that existed in the same layer by prior medical knowledge (more details see Appendix A.1), and then define the Tissue Graph $\mathcal{G}^{(T)} = (\mathcal{V}^{(T)}, \mathcal{E}^{(T)})$ with $N_t$ clinically essential tissue nodes and a global node, where $\mathcal{V}^{(T)}, \mathcal{E}^{(T)}$ denote the nodes and edges connecting them.

2) *Lesion Graph*: Following the conclusion that lesions diagnosed in the same tissue are more related (Zhang et al., 2020), we extract adequate

$<tissue,lesion>$ pairs from each sentence in the training corpus with artificial refining, and thereby relations in lesions can be summarized with tissues as a bridge (more details see Appendix A.2). Then, we define the Lesion Graph $\mathcal{G}^{(L)} = (\mathcal{V}^{(L)}, \mathcal{E}^{(L)})$ with $N_l$ lesion nodes and a global node, where $\mathcal{V}^{(L)}$, $\mathcal{E}^{(L)}$ denote the nodes and edges, respectively.

3) *Tissue-Lesion Graph*: Since Tissue and Lesion Graphs are fixed to learn stable semantics, Tissue-Lesion Graph is designed to depict detailed attributes' interconnections for each case. We modify the extracted $<tissue,lesion>$ pairs into $<tissue,relation,lesion>$ triplets, where the *relation* $\in \{0, 1\}$ denotes whether the tissue is paired with the lesion. For each tissue node $v_i^{(T)} \in \mathcal{G}^T$, its associated lesions can be denoted as a local Lesion Graph $\mathcal{G}^{(L_i)} \in \mathcal{G}^{(L)}$, and we represent the Tissue-Lesion Graph as $(\mathcal{G}^{(T)}, \{\mathcal{G}^{(L_i)}\})$ to store the inter-attribute correlations for each instance.

As an aggregation of three subgraphs, the PG encapsulates a wealth of medical knowledge. Next, we will explore how to represent this knowledge within the model.

### 3.2.2 Pathological Graph Embedding

Different from previous single-attribute learning methods (Zhang et al., 2020; Liu et al., 2021), we adopt an attribute-oriented graph convolution network in parallel for both Tissue and Lesion Graphs. Specifically, we first initialize the graph features by spatial visual features $V_g$ as (Zhang et al., 2020) to initially map each pathological node $v_i$ to different spatial regions, resulting in initialized tissue graph feature $T_f^0$ and lesion graph feature $L_f^0$. More

details are in Appendix A.4. Then, our graph convolution network follows Zhang et al. (2020); Liu et al. (2021) to propagate two graph features as:

$$T_f^{l+1} = update_1(T_f^l, msge_1(T_f^l, A_T)), \quad (2)$$
$$L_f^{l+1} = update_2(L_f^l, msge_2(L_f^l, A_L)), \quad (3)$$

where $T_f^l$, $L_f^l$ respectively denote the tissue and lesion graph features in the *l-th* layer, and $A_T, A_L$ are the normalized Laplacian of adjacency matrixes according to Tissue and Lesion Graph. $msge_1$, $msge_2$ are feature aggregation functions based on the adjacency matrix, while $update_1$ and $update_2$ propagate node features in PG.

After 2-layer propagation, we obtain the tissue graph embedding $T_f = \{T_{f_1}, ..., T_{f_{N_t}}\} \in \mathbb{R}^{N_t \times N \times d_g}$ and lesion graph embedding $L_f = \{L_{f_1}, ..., L_{f_{N_l}}\} \in \mathbb{R}^{N_l \times N \times d_g}$, where $T_{f_i} \in \mathbb{R}^{N \times d_g}$ and $L_{f_i} \in \mathbb{R}^{N \times d_g}$ respectively denote the *i-th* tissue and lesion node embedding. Afterward, we use a global average pooling on $T_f$ and $L_f$ to converge the information of $N$ scans and obtain the generalized node features of tissues $T_a \in \mathbb{R}^{N_t \times d_g}$ and lesions $L_a \in \mathbb{R}^{N_l \times d_g}$.

### 3.2.3 Pathological Graph Updating

Since the tissues and lesions are sparse and semantically correlated in Brain CT, node features in PG are supposed to be updated with the following concerns: 1) Concrete semantics of pathology groundings; 2) Tissue-lesion relations. Note that the updating is only the tuning of graph node features and does not involve any new connections within the tissue and lesion subgraphs.

To accurately represent the medical semantics of each node, we introduce an Intra-Attribute classification (IAC) module to dynamically update the node features. Separate IAC modules are employed for tissue and lesion nodes. Here, we take the tissue nodes as an example to illustrate IAC. We design a classification head, which takes $T_a$ as input, to predict the tissue labels $\hat{C}_t \in \mathbb{R}^{N_t}$. We then use BCE loss to optimize the IAC module as:

$$\mathcal{L}_{IAC_t} = -\sum_i^{N_t} W_{IAC_t}^i[C_t^i \log \hat{C}_t^i \quad (4)$$
$$+ (1 - C_t^i) \log(1 - \hat{C}_t^i)],$$

where $C_t \in \mathbb{R}^{N_t}$ denotes the ground tissue labels extracted from reports, $W_{IAC_t}$ is a weight matrix for tissue node updating, and $i$ denotes the current tissue index. Following this way, we can acquire the fine-grained pathology-grounded node features for both tissues and lesions.

While IAC captures the semantics of single attribute nodes, learning the inter-attribute relations between tissues and lesions is also essential, as our task is to generate reports with complex cranial findings rather than a simple classification. Therefore, we design an Inter-Correlation Alignment (ICA) module to match the correct tissue-lesion pairs via BCE loss. Specifically, we first extract the triplets in Tissue-Lesion Graph to depict tissue-lesion relations in a current case. As such, each sample is equipped with an attribute alignment matrix $M_a \in \{0, 1\}^{N_t \times N_l}$ as the ground-truth label, where the matrix element $e_{ij} = 1$ denotes the *i-th* tissue is associated with the *j-th* lesion in this sample. Afterward, we use dedicated node features ($T_a$ and $L_a$) to predict the inter-correlation matrix as:

$$\hat{M}_a = Sigmoid((T_a L_a^T) W_a^T + b_a) \in \mathbb{R}^{N_t \times N_l}, \quad (5)$$

where $W_a, b_a$ are parameters. The mutual information between two attributes is finally preserved by minimizing the following ICA loss:

$$\mathcal{L}_{ICA} = -\sum_i^{N_t} \sum_j^{N_l} W_{ICA_j}^i[M_{a_j}^i \log \hat{M}_{a_j}^i \quad (6)$$
$$+ (1 - M_{a_j}^i) \log(1 - \hat{M}_{a_j}^i)],$$

where $W_{ICA}$ is a learnable weight matrix, and $M_{a_j}^i$, $\hat{M}_{a_j}^i$ denote the (*i-th*,*j-th*) value in the ground-truth and predicted attribute matrix, respectively.

### 3.2.4 Cross-modal Contrastive Learning

Considering that detailed medical information is significant to find subtle pathological clues that depict critical medical conditions, the goal of our Contrastive Learning (CL) module is to represent the alignment of fine-grained pathological semantics from visual and textual modalities. To achieve this, we regard the nodes learned by IAC and ICA as fine-grained visual semantics since they comprehensively considered the intra- and inter-attribute knowledge and focused on salient pathological regions. Then, we extract textual features of corresponding words in the report via the shared textual embedding layers in textual decoder and map each node feature to its matched textual embeddings by contrastive learning. Specifically, we use tissue nodes for example. Given the *j-th* sample with $N$

| Methods | B1 | B2 | B3 | B4 | M | RG | C | P | R | F1 |
|---|---|---|---|---|---|---|---|---|---|---|
| Show-Tell(Vinyals et al., 2015)[†] | 37.2 | 25.5 | 18.0 | 12.9 | 25.8 | 35.6 | 16.6 | 51.4 | 53.4 | 49.0 |
| Soft ATT(Xu et al., 2015)[†] | 42.0 | 28.4 | 19.7 | 13.9 | 27.6 | 35.1 | 18.5 | 51.8 | 65.5 | 55.1 |
| HRNN(Krause et al., 2017)[†] | 39.0 | 26.3 | 18.3 | 13.0 | 28.8 | 34.7 | 16.3 | **53.6** | 37.7 | 40.6 |
| Up-Down(Anderson et al., 2018)[†] | 40.7 | 27.6 | 19.1 | 13.5 | 27.0 | 35.4 | **20.1** | 51.9 | 63.3 | 54.5 |
| MRMA(Xue et al., 2018)[†] | 39.7 | 26.9 | 18.8 | 13.4 | **29.0** | 34.9 | 15.6 | 53.0 | 59.0 | 52.7 |
| R2Gen(Chen et al., 2020)[†] | 43.1 | 27.5 | 18.3 | 12.4 | 28.1 | 34.1 | 15.8 | 52.2 | 57.3 | 51.9 |
| WCL(Yan et al., 2021)[†] | 42.6 | 28.6 | 19.9 | 14.1 | 27.4 | 34.9 | 17.5 | 52.3 | 60.9 | 52.7 |
| R2Gen-CMN(Chen et al., 2021)[†] | 40.1 | 25.1 | 16.0 | 10.3 | 26.2 | 32.3 | 13.9 | 51.1 | 50.2 | 46.5 |
| CMMRL(Qin and Song, 2022)[†] | 33.5 | 19.5 | 11.7 | 7.3 | 22.3 | 28.5 | 10.0 | 49.3 | 45.4 | 44.0 |
| XProNet(Wang et al., 2022)[†] | 40.1 | 27.1 | 18.9 | 13.2 | 26.1 | 35.2 | 17.9 | 52.0 | 61.3 | 53.8 |
| WGAM(Yang et al., 2021)[†] | 43.6 | 29.3 | 20.4 | 14.5 | 27.9 | 35.1 | 18.3 | 52.9 | 66.5 | 55.9 |
| Ours | **45.0** | **30.8** | **21.6** | **15.5** | 28.7 | **36.5** | 19.9 | **53.6** | **67.7** | **57.2** |

Table 1: The performance of our PGCA compared with previous state-of-the-art models on the Brain CT report generation dataset BCT-CHR. The best results are highlighted in bold. † denotes our re-implementation results.

scans and a report, we first project node features and associated word embeddings into a latent space, resulting in $V_{tj} = \{V_{t_j}^{(1)}, V_{t_j}^{(2)}, ..., V_{t_j}^{(N_{t_j})}\}$ and $R_{tj} = \{R_{t_j}^{(1)}, R_{t_j}^{(2)}, ..., R_{t_j}^{(N_{t_j})}\}$, where $V_{t_j}^{(i)} \in \mathbb{R}^{d_g}$, $R_{t_j}^{(i)} \in \mathbb{R}^{d_g}$ denote the projected visual and textual features, and $N_{t_j}$ is the number of tissues reported in the *j-th* sample. We then use the loss function similar to temperature-normalized InfoNCE (van den Oord et al., 2018) loss for visual-textual alignment:

$$\mathcal{L}_{CL_t} = -\sum_{u=1}^{N_{t_j}} \log \frac{\exp(\frac{s(V_{t_j}^{(u)}, R_{t_j}^{(u)})}{\tau})}{\sum_{v=1, u \neq v}^{N_{t_j}} \exp(\frac{s(V_{t_j}^{(u)}, R_{t_j}^{(v)})}{\tau})},$$

(7)

where $s(.)$ measures the cosine similarity between cross-modal vectors, and $\tau$ is the temperature hyperparameter. Following the same process of tissue semantic alignment, we can also obtain the lesion contrastive learning loss $\mathcal{L}_{CL_l}$.

### 3.3 Overall objective function

Finally, our model is optimized by the total loss regarding the report generation branch and PGCA branch, which is defined as:

$$\mathcal{L} = \mathcal{L}_g + \lambda_1 \mathcal{L}_{IAC} + \lambda_2 \mathcal{L}_{ICA} + \lambda_3 \mathcal{L}_{CL}, \quad (8)$$

where $\mathcal{L}_{IAC}$ and $\mathcal{L}_{CL}$ are calculated by the sum of tissue and lesion aspects. $\lambda_1$, $\lambda_2$, and $\lambda_3$ are the coefficients to balance the total loss.

## 4 Experiments

### 4.1 Dataset

We evaluate our model on the Brain CT report generation dataset BCT-CHR (Yang et al., 2021).

There are 49,152 CT scans and 2048 Chinese reports from 2048 anonymous samples, and each sample includes 24 scans over multiple pathological layers for various abnormal detection and a paired patient report. Following Yang et al. (2021), the dataset is split by 7 : 2 : 1 for training, testing, and validation, respectively. When tokenizing reports, the words with less than 2 occurrences are dropped, counting to 798 words in the vocabulary. Notably, the English translations are only for better understanding and are not used in training.

### 4.2 Evaluation Metrics

We fully evaluate the performance on the NLG (Natural Language Generation) and CE (Clinical Evaluation) metrics. NLG metrics contains BLEU (Papineni et al., 2002), METEOR (Lavie and Agarwal, 2007), ROUGE-L (Lin, 2004) and CIDEr (Vedantam et al., 2015), which are denoted as B1, B2, B3, B4, M, RG, and C. Then, according to the pathological knowledge obtained by radiologists, we use 24 keywords ("basal ganglia", "edema", "lateral ventricle", etc.) to evaluate the CE metrics: P (Precision), R (Recall), and F1 score.

### 4.3 Implementation Details

We set the scan number $N = 24$ for each sample and reshape the size of scans to $512 \times 512$. The scan features are extracted by ResNet101 (He et al., 2016), which is pretrained on ImageNet (Deng et al., 2009) and fine-tuned on CQ500 dataset (Chilamkurthy et al., 2018). The hyperparameters are tuned via the validation set. Empirically, the loss coefficients $\{\lambda_1, \lambda_2, \lambda_3\}$, temperature value $\tau$ and graph node dimension $d_g$ are set to $\{0.3, 0.001, 0.2\}$, 0.4 and 512, respectively. We

| Methods | Attributes | | Module Loss | | | B1 | B2 | B3 | B4 | M | RG | C |
|---|---|---|---|---|---|---|---|---|---|---|---|---|
| | tissue | lesion | $\mathcal{L}_{IAC}$ | $\mathcal{L}_{ICA}$ | $\mathcal{L}_{CL}$ | | | | | | | |
| Baseline | ✗ | ✗ | ✗ | ✗ | ✗ | 40.7 | 27.5 | 19.2 | 13.8 | 26.4 | 35.3 | 16.3 |
| (a) | ✗ | ✗ | ✗ | ✗ | ✓ | 41.8 | 28.4 | 19.9 | 14.2 | 27.1 | 35.7 | 18.7 |
| (b) | ✓ | ✗ | ✓ | ✗ | ✗ | 42.5 | 28.9 | 20.3 | 14.6 | 27.3 | 35.7 | 18.0 |
| (c) | ✓ | ✗ | ✓ | ✗ | ✓ | 43.5 | 29.6 | 20.7 | 14.9 | 27.8 | 35.9 | 19.7 |
| (d) | ✓ | ✓ | ✓ | ✗ | ✗ | 43.9 | 29.7 | 20.8 | 14.8 | 28.3 | 36.0 | **21.3** |
| (e) | ✓ | ✓ | ✓ | ✗ | ✓ | 44.1 | 29.8 | 20.8 | 14.8 | 27.8 | 35.9 | 19.3 |
| Ours | ✓ | ✓ | ✓ | ✓ | ✓ | **45.0** | **30.8** | **21.6** | **15.5** | **28.7** | **36.5** | 19.9 |

Table 2: Ablation studies of our proposed method. The **Baseline** model is an encoder-decoder framework with attention mechanism that is similar to (Yang et al., 2021). **(a)**, **(b)**, **(c)**, **(d)** and **(e)** respectively denote the adding of different pathological graph knowledge (i.e. tissue and lesion) and module losses.

train the model with Adam optimizer (Kingma and Ba, 2015) on an NVIDIA RTX 3090 GPU with a batch size of 4 for 60 epochs. The learning rate is initially set to 4e-4 in the first 30 epochs, and we decrease it by a 0.8 rate per 3 epochs after that.

## 4.4 Results and Discussion

### 4.4.1 Comparison Study

Besides the only Brain CT report generation method **WGAM** (Yang et al., 2021), we also reproduce some SOTA models in image captioning (**Show-Tell** (Vinyals et al., 2015), **Soft ATT** (Xu et al., 2015), **HRNN** (Krause et al., 2017), **Up-Down** (Anderson et al., 2018)) and chest X-ray Report generation (**MRMA** (Xue et al., 2018), **R2Gen** (Chen et al., 2020), **WCL** (Yan et al., 2021), **R2Gen-CMN** (Chen et al., 2021), **CMMRL** (Qin and Song, 2022), **XProNet** (Wang et al., 2022)) on BCT-CHR dataset for comprehensive comparisons.

As shown in Table 1, our method outperforms the competitors on almost all evaluation metrics. Specifically, the negative results in **Show-Tell** indicate that complex medical semantics may not be fully captured without an efficient cross-modal interaction. Benefit from the cross-modal attention (**Soft ATT**, **Up-Down**, and **WGAM**) and instance-level contrastive learning (**WCL**), the models gain improving results. Unexpectedly, by simply using memory vectors without domain knowledge, **R2Gen-CMN** and **CMMRL** perform poorly on most metrics, while **XProNet** adds class-related knowledge and gets better results. This reflects the effect of knowledge and the difficulties of our task. **WGAM** benefits from weakly-guided attention to capture visual features of Brain CT and achieves higher BLEU scores. Despite these successes, detailed visual and textual features are not sufficiently learned in MRG systems, which fails to generate accurate Brain CT reports. With the use of pathologi-

cal graph and fine-grained cross-modal alignment, our PGCA achieves the best performance in contrast with previous arts. Especially, compared with the only work **WGAM** for Brain CT report generation, PGCA gains remarkable improvement in B3 (20.4% → 21.6%), RG (35.1% → 36.5%), and F1 (55.9% → 57.2%), which justifies that PGCA does not confuse the learning of some essential pathological details, but generating sentences with more accurate topics. It is worth noting that PGCA is slightly inferior on CIDEr, which is more sensitive to word frequency. The reason may be that our graph contains some high-frequency terms, leading to more occurrences of them in generated reports and slightly decreasing the CIDEr.

### 4.4.2 Ablation Study

Table 2 summarizes the results of ablation studies to verify the contributions of graph knowledge injection and fine-grained contrastive learning. We remove the supervision of attention in **WGAM** as our Baseline, and then progressively add attribute components (*tissue* and *lesion*) and the module losses ($\mathcal{L}_{IAC}$, $\mathcal{L}_{ICA}$ and $\mathcal{L}_{CL}$), respectively denoting the incorporation of two types of attribute knowledge in pathological graph and the utilization of proposed modules (i.e. IAC, ICA, and CL).

As shown in Table 2, the advantages of adding tissue and lesion attributes can be well reflected by the improvement from Baseline to (b) and further to (d), which justifies our assumption for learning subtle tissues and lesions. The comparison between (b) and (c) also indicates the contributions brought by fine-grained contrastive learning. Note that, (a) denotes the Baseline model with the incorporation of CLIP loss (Radford et al., 2021) that directly uses image-text pairs for coarse-grained contrastive learning. We can observe that, only with fine-grained tissues for detailed feature align-

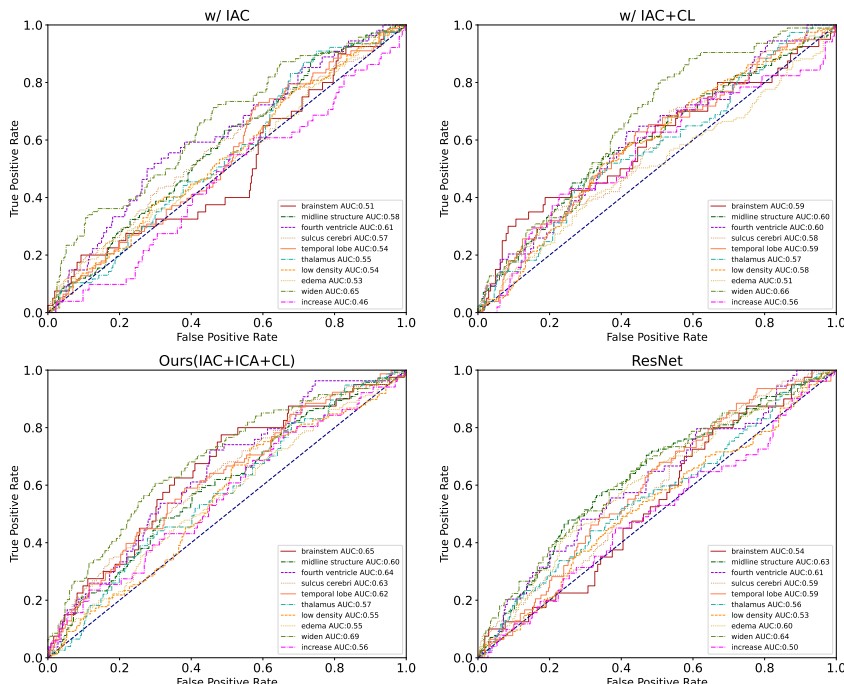

Figure 3: Comparison of pathological elements classification with different model settings and classic ResNet model. The left-top chart shows the ROC curves of IAC module without the effect of other modules, and each curve denotes one pathological element. Based on the first model, CL and ICA modules are gradually added and corresponded to the following charts. The last chart shows the classification performance of classic ResNet.

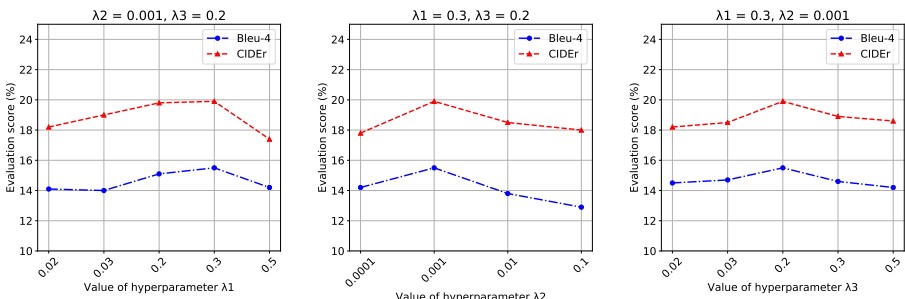

Figure 4: Parameter sensitivity tests for the loss function. Each hyperparameter is evaluated in a specific range, with the other two kept at their optimal values.

ment, (c) has already surpassed (a) by a large margin, and when the lesion attribute is considered, (e) gains further improvements. This indicates the effectiveness of our fine-grained cross-modal feature alignment idea in MRG. Compared with (d), (e) gains inferior results on METEOR, ROUGE and CIDEr, the reason may be that pathological semantics in two attributes are separately learned without a clinically important interaction. With the incorporation of ICA, our final model achieves the best performance. These results verify the capabilities of our modules to boost report generation.

To further confirm the module contributions for pathology identification, in Figure 3, we visualize the ROC curves for the classification of 10 essential pathological elements summarized by experienced physicians. Since classification is only performed by our IAC module, the left chart shows the performance of the model only with IAC and the report generation branch. With the progressive addition of CL and ICA modules, the model gains better performance, which further indicates that CL and ICA can help enhance the node feature representation and boost the recognition of pathological elements. Besides, we also compare the classification performance with the classic ResNet101 (He et al., 2016) model. It is noticeable that our model outperforms the ResNet by a large margin, which not only indicates the effectiveness of our graph learning and cross-modal alignment methods but

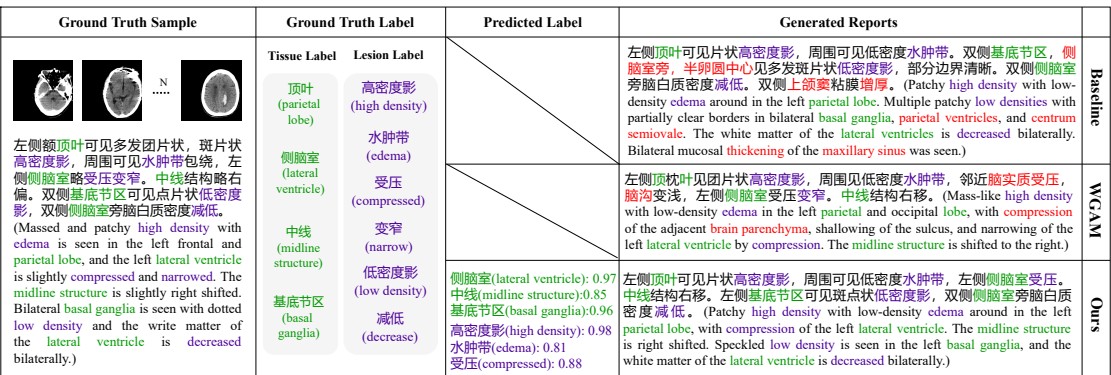

Figure 5: Example of reports generated by the Baseline, WGAM, and our proposed model. The correct tissues, lesions, and error-predicted pathological elements are marked in green, purple and red, respectively. English translations of the Chinese reports are given for better understanding.

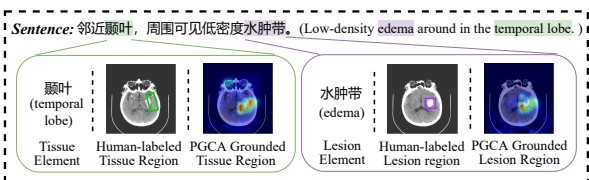

Figure 6: Visualization of fine-grained pathology recognition. Given a sentence, Grad-CAM is used to ground salient visual regions of pathological elements.

also hints report generation and classification can boost each other.

To examine the impact of hyperparameters of loss (*i.e.*, $\lambda_1$, $\lambda_2$ and $\lambda_3$ in Eq. 8) on our model, we provide parameter sensitivity experiments in Figure 4, where we fix two optimal hyperparameters and vary the third within a specific range. We can observe that too low or too high values of hyperparameters decrease the performance. The reason may be because the too low score of $\lambda_1$, $\lambda_2$, and $\lambda_3$ respectively lead to inadequate pathological semantic learning, inter-attribute matching, and visual-textual alignment, which degrades the feature representation. Too high $\lambda_1$, $\lambda_2$, and $\lambda_3$ may weaken the training of report generation-related parameters and cause sub-optimal results.

### 4.4.3 Qualitative Analysis

Figure 5 visualizes the generated reports from a qualitative perspective, we list the ground truth image-report pairs with pathological labels, predicted labels generated by IAC module, and the reports generated by Baseline, WGAM, and our PGCA. It is observed that Baseline and WGAM suffer from generating suboptimal reports with inaccurate cranial tissues and wrong-matched lesions. With the incorporation of our enriched graph knowledge and fine-grained cross-modal feature alignment, PGCA can not only recognize complex relations of pathologies and correctly predict labels, but also generate high-quality reports based on the understanding of pathological elements. We also visualize the ability to recognize fine-grained pathology semantics in Figure 6, the salient regions grounded by our IAC module are generally consistent with human-labeled boxes. These qualitative analyses further demonstrate the superiority of our PGCA model.

## 5 Conclusion

We propose a Pathological Graph-driven Cross-modal Alignment model for Brain CT report generation. First, we construct a Pathological Graph that incorporates detailed medical knowledge, allowing for the injection of this knowledge into dedicated node features through graph embedding and updating, thereby achieving fine-grained visual representations. Second, we align the learned node features with the corresponding word embeddings by cross-modal contrastive learning, to further boost report generation. Extensive experiments demonstrate that our model achieves superior performance in generating clinically accurate reports.

## Limitations

This paper is mainly toward Brain CT medical report generation and may not generalize well to other medical imaging, such as chest datasets (e.g. MIMIC-CXR and IU-Xray) and ophthalmology datasets (e.g. FFA-IR), without further adaption. Besides, we only focus on mining detailed tissue and lesion elements in brain findings and building fine-grained visual-textual alignment based

on them, which lacks the consideration of more specific details like tissue orientation or lesion size. In the future, explorations of how to incorporate useful medical knowledge with different types and granularity into the medical report generation model will substantially contribute to this field.

## Ethics Statement

The medical images in the BCT-CHR dataset have undergone a thorough process of de-identification to protect participants' privacy. Additionally, we have obtained the required data permissions, ensuring our work meets ethical standards.

## Acknowledgements

This work was supported in part by the National Natural Science Foundation of China under Grant 61906007, 62276010, and 62306253, in part by the R&D Program of Beijing Municipal Education Commission under Grant KM202110005022 and KZ202210005009.

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

## A  Appendix

### A.1  Details of Tissue Graph Construction

In the Tissue Graph, nodes are defined as $N_t$ tissue terms, which are summarized in two aspects: 1) The knowledge and expertise of experienced radiologists; 2) The frequency of tissue words in the training corpus. We present these tissues in Table 3. Then, with the given nodes, a rule-based algorithm is designed to assign the connecting edges via the prior domain knowledge, i.e. tissues in the same scan layer are more related, otherwise not. Inspired by Li et al. (2022), our algorithm contains the following three steps:

1) **Knowledge Formatting**: We format the expert knowledge into $<layer,tissue>$ pairs, where $layer$ indicates a layer name (e.g. "canthus earline layer", "suprasellar cistern layer", "upper cerebral cortex layer", etc.), and $tissue$ denotes the tissue that can be observed in this layer.

2) **Dependency Parsing**: With the formatted knowledge pairs, for each layer, we collect a set of paired tissues and then parse the dependency of these tissues into $<tissue_1,tissue_2>$ pairs, where $tissue_1$, $tissue_2$ denotes two different tissues paired with the same layer. We traverse all the layers and finally obtain a large set of tissue dependency pairs.

3) **Entity Linking**: A matrix $E_{t_0} \in \mathbb{R}^{N_t \times N_t}$ is set to store the relation of $N_t$ tissue nodes, which is initialized as an identity matrix. Then, we traverse the extracted tissue dependency pairs $<tissue_1,tissue_2>$, and add the weight by 0.2 to update the bidirectional edges of $tissue_1$ and $tissue_2$ for each time. If the edge between two different tissues reaches 0.8, we stop to add it. Finally, we add a column and a row with the weight of 1.0 to depict the fully connected edges of a global node, resulting in the edge matrix $E_t \in \mathbb{R}^{(N_t+1) \times (N_t+1)}$.

In total, the Tissue Graph contains 133 tissue dependency pairs, and more information are listed in Table 4.

### A.2  Details of Lesion Graph Construction

We comprehensively combine the expert opinions and the word frequency in the training corpus to extract $N_l$ lesion terms as our lesion nodes, which are presented in Table 5. Similar to the linking of tissue nodes, we use the same rule-based algorithm to build lesion edges. Different from the knowledge utilized in constructing the Tissue Graph, here we follow the assumption of Zhang et al. (2020): le-

| ID | Tissue |
|----|--------|
| 1 | 侧脑室(lateral ventricle) |
| 2 | 丘脑(thalamus) |
| 3 | 基底节区(basal ganglia) |
| 4 | 脑干(brainstem) |
| 5 | 顶叶(parietal lobe) |
| 6 | 颞叶(temporal lobe) |
| 7 | 额叶(frontal lobe) |
| 8 | 枕叶(occipital lobe) |
| 9 | 脑室(ventricle) |
| 10 | 中线(midline structure) |
| 11 | 第三脑室(third ventricle) |
| 12 | 第四脑室(fourth ventricle) |
| 13 | 蝶窦(sphenoid sinus) |
| 14 | 筛窦(ethmoid sinus) |
| 15 | 上颌窦(maxillary sinus) |
| 16 | 颞枕叶(temporal occipital lobe) |
| 17 | 半卵圆中心(centrum semiovale) |
| 18 | 脑实质(brain parenchyma) |
| 19 | 脑沟(sulcus cerebri) |

Table 3: Components of tissue terms in our Tissue Graph, which are summarized by the knowledge of radiologists and the word frequency. English translations of the Chinese words are given for better understanding.

|  | Number |
|--|--------|
| Tissue | 19 |
| Layer | 8 |
| $<layer,tissue>$ | 61 |
| $<tissue_1,tissue_2>$ | 133 |

Table 4: Statistics of our Tissue Graph, containing the number of tissue entities, layer categories, $<layer,tissue>$ pairs, and $<tissue_1,tissue_2>$ pairs.

sions diagnosed in the same tissue are more related, otherwise not.

In **Knowledge Formatting**, we extract adequate $<tissue,lesion>$ pairs from each sentence in the training corpus, and manually adjust the pairs to ensure correctness. In **Dependency Parsing**, we parse the dependencies of lesions into $<lesion_1,lesion_2>$, where $lesion_1$, $lesion_2$ denotes two different lesions paired with the same tissue. In **Entity Linking**, we store the relation of $N_l$ lesion nodes in $E_{l_0} \in \mathbb{R}^{N_l \times N_l}$, which is initialized as an identity matrix. We traverse the collected $<lesion_1,lesion_2>$ pairs and add the weight by 0.1 to update the bidirectional edges of $lesion_1$ and $lesion_2$ for each time until the score reaches 0.9. Finally, a column and a row with the weight of 1.0 are added to represent the fully connected edges

## Case (I) for Brain CT report generation

| Ground Truth Sample | Ground Truth Label | | Predicted Label | Generated Reports | |
|---|---|---|---|---|---|
| | **Tissue Label** | **Lesion Label** | | | |
| 左侧基底节区见团块状高密度影，周边可见少许低密度区，左侧侧脑室受压变窄。中线略右移。 (Massed high density with a few low-density areas are seen in the left basal ganglia, and the left lateral ventricle is compressed and narrowed. The midline structure is slightly right-shifted. ) | 基底节区 (basal ganglia) | 高密度影 (high density) | | 左侧额叶见团片状高密度影，周围见低密度水肿带。双侧基底节区见点状低密度影，双侧侧脑室旁脑白质密度减低。(Mass-like high density with low-density edema around the left frontal lobe. Point-like low density is seen in the bilateral basal ganglia. The white matter of the lateral ventricles is decreased bilaterally.) | Baseline |
| | 侧脑室 (lateral ventricle) | 受压 (compressed) | | 左侧颞叶见团片状高密度影，周围见低密度水肿带，邻近脑实质受压，脑沟变浅。双侧侧脑室旁，半卵圆中心见多发点状低密度影，双侧侧脑室旁脑白质密度减低。 (Mass-like high density with low-density edema in the left temporal lobe, with compression of the adjacent brain parenchyma, and shallowing of the sulcus cerebri. Point-like low density is seen in the bilateral basal ganglia, lateral ventricle, and centrum semiovale, and the white matter of the lateral ventricle is decreased bilaterally.) | WGAM |
| | 中线 (midline structure) | 变窄 (narrow) | 基底节区(basal ganglia):0.96 侧脑室(lateral ventricle): 0.97 高密度影(high density): 0.98 受压(compressed): 0.93 | 左侧基底节区见团片状高密度影，周围见低密度水肿带，左侧侧脑室受压变窄。中线结构略右移。双侧基底节区见多发点片状低密度影，部分边界清晰。双侧侧脑室旁脑白质密度减低。(Massed high density with low-density edema around the left basal ganglia and the left lateral ventricle is compressed and narrowed. The midline structure is slightly right-shifted. Multiple point-like low densities with partially clear borders in the bilateral basal ganglia. The white matter of the lateral ventricle is decreased bilaterally.) | Ours |

## Case (II) for Brain CT report generation

| Ground Truth Sample | Ground Truth Label | | Predicted Label | Generated Reports | |
|---|---|---|---|---|---|
| | **Tissue Label** | **Lesion Label** | | | |
| 左侧顶叶可见片状高密度影，周围见带状低密度影。中线略右移。基底节区，双侧侧脑室旁多发斑点，斑片状低密度影，双侧侧脑室旁脑白质密度减低。 (Patchy high-density with band-like low-density around the left parietal lobe. The midline structure is slightly right shifted. Speckled, patchy low density is seen in the basal ganglia and bilateral lateral ventricles, and the white matter of the lateral ventricle is decreased bilaterally.) | 顶叶 (parietal lobe) | 高密度影 (high density) | | 左侧顶叶可见片状高密度影，周围可见低密度水肿带。双侧基底节区，侧脑室旁，半卵圆中心见多发斑点状低密度影，部分边界清晰。双侧侧脑室旁脑白质密度减低。双侧上颌窦粘膜增厚。 (Patchy high density with low-density edema around the left parietal lobe. Multiple patchy low densities with partially clear borders in bilateral basal ganglia, parietal ventricles, and centrum semiovale. The white matter of the lateral ventricles is decreased bilaterally. Bilateral mucosal thickening of the maxillary sinus was seen.) | Baseline |
| | 中线 (midline structure) | 低密度影 (low density) | | 左侧顶叶可见片状高密度影，周围可见低密度水肿带，邻近脑实质受压，脑沟变浅，左侧侧脑室受压变窄。中线结构右移。 (Patchy high density with low-density edema in the left parietal lobe, with compression of the adjacent brain parenchyma, shallowing of the sulcus cerebri, and narrowing of the left lateral ventricle by compression. The midline structure is right shifted.) | WGAM |
| | 基底节区 (basal ganglia) | 减低 (decrease) | 基底节区(basal ganglia):0.96 侧脑室(lateral ventricle): 0.97 高密度影(high density): 0.98 减低(decrease): 0.89 | 左侧顶叶可见片状高密度影，周围可见低密度水肿带，左侧侧脑室受压。中线结构右移。左侧基底节区可见斑点状低密度影，双侧侧脑室旁脑白质密度减低。 (Patchy high density with low-density edema around the left parietal lobe, with compression of the left lateral ventricle. The midline structure is right-shifted. Speckled low density is seen in the left basal ganglia, and the white matter of the lateral ventricle is decreased bilaterally.) | Ours |
| | 侧脑室 (lateral ventricle) | | | | |

Figure 7: More cases of reports generated by the Baseline, WGAM, and our proposed model. English translations of the Chinese reports are given for better understanding.

of one global node, resulting in the edge matrix $E_l \in \mathbb{R}^{(N_l+1)\times(N_l+1)}$.

In total, our Lesion Graph includes 652 lesion dependency pairs, and more details are shown in Table 6.

| | Number |
|---|---|
| Lesion | 13 |
| Tissue | 19 |
| $<tissue,lesion>$ | 159 |
| $<lesion_1,lesion_2>$ | 652 |

Table 6: Statistics of our Lesion Graph, containing the number of lesion entities, associated tissues, $<tissue,lesion>$ pairs, and $<lesion_1,lesion_2>$ pairs.

### A.3 More Cases of Brain CT Report Generation

More cases of Brain CT report generation are presented in Figure 7. The correct tissues, lesions, and error-predicted pathological elements are marked in green, purple and red, respectively.

### A.4 Details of Graph Features Initialization

To map the graph features into pathology-related spatial regions, we follow the feature initialization method in Zhang et al. (2020) with some modifications. Given the spatial visual feature $V_g \in \mathbb{R}^{(N*H)\times d_g}$ ($N = 24$, $H = 196$, $d_g = 512$), where $N$, $H$, $d_g$ denote the number of scans, grids,

| ID | Lesion |
|---|---|
| 1 | 高密度影(high density) |
| 2 | 低密度影(low density) |
| 3 | 受压(compressed) |
| 4 | 增宽(widen) |
| 5 | 增高(increase) |
| 6 | 增厚(thicken) |
| 7 | 水肿带(edema) |
| 8 | 减低(decrease) |
| 9 | 密度影(density shadow) |
| 10 | 移位(shift) |
| 11 | 变窄(narrow) |
| 12 | 左移(left shift) |
| 13 | 肿胀(swell) |

Table 5: Components of lesion terms in our Lesion Graph, which are summarized by the knowledge of physicians and the word frequency. English translations of the Chinese words are given for better understanding.

and channels, we duplicate it twice for initializing tissue and lesion graph features separately. We mainly take the tissue graph feature initialization for example.

First, we transform the channel dimension of $V_g$ into tissue number $N_t$ as:

$$
\begin{aligned}
V_{tg1} &= Relu(V_g W_{t_1}^T + b_{t_1}), & (9) \\
V_{tg2} &= Relu(V_{tg1} W_{t_2}^T + b_{t_2}), & (10) \\
V_{tg3} &= Relu(V_{tg2} W_{t_3}^T + b_{t_3}), & (11)
\end{aligned}
$$

where Relu(.) denotes the ReLU activation function. $W_{t_1} \in \mathbb{R}^{(N*H)\times512}$, $W_{t_2} \in \mathbb{R}^{(N*H)\times128}$, $W_{t_3} \in \mathbb{R}^{(N*H)\times N_t}$ are learnable weights. $b_{t_1}$, $b_{t_2}$, $b_{t_3}$ are learnable biases. We use the transformation result $V_{tg3} \in \mathbb{R}^{(N*H)\times N_t}$ to represent the graph information.

Then, to bind the graph information $V_{tg3}$ with specific visual regions, we perform the multiplication of matrices $V_g$ and $V_{tg3}$ along the dimension of spatial grids, resulting in $V_{tg} \in \mathbb{R}^{N \times H \times N_t}$. In this way, each row of $V_{tg}$ can depict one specific tissue region, and we represent it as one tissue node feature. Afterward, we conduct average pooling on the dimension of node number $N_t$, to obtain the global node feature $V_{tglobal} \in \mathbb{R}^{N \times H}$ for the tissue graph, which contains general node representations. Finally, we concat $V_{tg}$ and $V_{tglobal}$, and reshape it as the initialized tissue graph feature $T_f^0 \in \mathbb{R}^{(N_t+1)\times(N*H)}$.

As the same process, the lesion graph feature is initialized as $L_f^0 \in \mathbb{R}^{(N_l+1)\times(N*H)}$, where $N_l$ is the number of lesion nodes.