# OpenReview forum: "Granularity Matters: Pathological Graph-driven Cross-modal Alignment for Brain CT Report Generation"
_EMNLP/2023/Conference — EMNLP 2023 Main_

### Official Review · Reviewer_kj9V · 2023-07-22

**Soundness:** 4

**Excitement:**

4: Strong: This paper deepens the understanding of some phenomenon or lowers the barriers to an existing research direction.

**Paper Topic And Main Contributions:**

This paper proposes a novel Pathological Graph-driven Cross-modal Alignment (PGCA) model for accurate and robust Brain CT report generation.

The main contributions are as follows:
1. The paper proposes a novel framework to seamlessly capture detailed domain knowledge from Pathological Graph, and explicitly align fine-grained visual and textual features of pathology.
2. The paper, for the first time, introduce the idea of decoupling cranial tissues and lesions via Pathological Graph into the medical report generation area, which is capable of handling fine-grained alignment between long-text report and multiple scans.
3. The paper comprehensively validate the proposed model on the BCT-CHR dataset. The experimental results indicate that the proposed method surpasses previous works in medical report generation.

**Questions For The Authors:**

A) Figure 1, all  graph edges between tissue and lesion are 1.0. Why is it so? Also, why all lesion edges are 0.9? Is it true that they are always the case? It is unclear how you constructed the edge probability. Parts of the Appendix may be needed in the main text for explaining.
B) Figure 2, the ground truth report should be used for calculating the generative loss. It is weird that it has no connection between the (a) and ground truth repot.
C) Figure 2, the textual embedding on the right is claimed to be shared with the textual decoding at the bottom. There is no description about how it is shared. The right embedding layer should be encoding layer.  How can it be  shared with the decoding layer?
D) Figure 3, what does the numbers (e.g.: 0.65) next to the line name means?
E) Based on the models structure (Figure 2) and the description, it is unclear to me which part is outputting the 'predicted labels' in  Figure 4.
F) The loss coefficients are set to 0.3, 0.001 and 0.2. Why is it so? Why as low as 0.001?
G) Table 1,  why there is a line set between up-down and MRMA? Are they belonging to two different groups of models?

**Reasons To Accept:**

The proposed Pathological Graph is novel as it decouples cranial tissues  and lesions. The experiments well demonstrates the benefits of using such graph. Besides that, it also helps the model interpretability of models.

**Reasons To Reject:**

There is only one comparison method relating to brain CT report generation and this method WGAM is from 2 years ago. There are possibly unlisted related works on this task. In addition, it is uncleared that, models other than WGAM are from medical domain or not.  Considering the fact that brain CT is has its unique challenges comparing to other medical images and other natural images, the experiment result may not be as valid as the paper claims.

**Reproducibility:**

4: Could mostly reproduce the results, but there may be some variation because of sample variance or minor variations in their interpretation of the protocol or method.

**Reviewer Confidence:**

3: Pretty sure, but there's a chance I missed something. Although I have a good feel for this area in general, I did not carefully check the paper's details, e.g., the math, experimental design, or novelty.

---

> ### Author Rebuttal · Authors · 2023-08-28
>
> **Q1. There is only one comparison method WGAM (from 2 years ago) relating to brain CT report generation. There are possibly unlisted related works on this task. It is also unclear whether the compared models other than WGAM are from medical domain or not. Considering the fact that brain CT has its unique challenges comparing to other medical images and other natural images, the experiment result may not be as valid as the paper claims.**
>
> **A1.** We appreciate your concern regarding the comparison with only one brain CT report generation method, WGAM. We acknowledge that brain CT report generation is an underexplored area, and WGAM is currently the only available method for this task. Our paper provides comprehensive coverage of this emerging field, and our study contributes by being at the forefront of this evolving domain.
>
> To ensure the effectiveness of our method, we have incorporated comprehensive comparisons with SOTA Chest X-Ray report generation (medical domain) and image captioning (non-medical domain) models in addition to WGAM. Please refer to L470-480 for more details. All these comparison methods share similar visual-language generation tasks to brain CT report generation and have been commonly used for comparison purposes in the report generation area.
>
> In experiments, all the comparison models were reproduced and trained on the brain CT dataset with consistency to ours to overcome the unique challenges of this task. Specifically, the comparison models underwent the same visual feature extracting and fine-tuning process as our baseline. We also standardized the training epochs at 60 and selected the best outcomes from 5 runs, maintaining consistency and ensuring fairness and reliability in our comparative analysis.
>
>
>
>
> **Q2. Figure 1, all graph edges between tissue and lesion are 1.0. Why is it so? Also, why all lesion edges are 0.9? Is it true that they are always the case? It is unclear how you constructed the edge probability. Parts of the Appendix may be needed in the main text for explaining.**
>
> **A2.** Thanks for your constructive suggestion. (1) The edges within the tissue-lesion graph are binary (0 or 1), signifying whether a specific tissue corresponds to a particular lesion in the report. In Figure 1, we have displayed only the edges with a value of 1 for conciseness. (2) Notably, lesion edge values range from 0 to 1, with the instances of 0.9 being specific occurrences. To enhance clarity and avoid misinterpretation, we will adjust the cases illustrated in Figure 1 accordingly. (3) The construction of edge probabilities involves a comprehensive approach. Through knowledge formatting and dependency parsing, relation pairs are identified (e.g. 133 <tissue, tissue> pairs for the tissue graph, and 652 <lesion, lesion> pairs for the lesion graph). These pairs guide the addition of weight to update bidirectional edges for corresponding tissues or lesions until a predetermined upper limit is reached. The detailed methodology can be found in Appendix A.1, A.2. For clarity, more information about building the edge probabilities will be added in the main text.
>
> **Q3. Figure 2, the ground truth report should be used for calculating the generative loss. It is weird that it has no connection between the (a) and ground truth report.**
>
> **A3.** Thanks for the comment. We apologize for any confusion caused by the figure. In Figure 2(a), the generated report is indeed compared to the ground truth report by default, and the generative loss is calculated accordingly. To make this clearer, we will modify Figure 2 in the final version of our manuscript to better illustrate the connection between the generated report and the ground truth report.
>
> **Q4. Figure 2, the textual embedding on the right is claimed to be shared with the textual decoding at the bottom. There is no description about how it is shared. The right embedding layer should be encoding layer. How can it be shared with the decoding layer?**
>
> **A4.** Thanks for your valuable comment. Actually, the textual decoder in Figure 2 contains two parts: a "textual embedding" layer and a language generation model. The "textual embedding" layer embeds words in decoder and is shared with the "right embedding layer". In our initial manuscript, we did not divide the textual decoder to keep it concise, which may have caused confusion. We will refine Figure 2 in the final version to provide a clearer visualization of the weight-sharing nature and structure of the textual decoder.
>
> **Q5. Figure 3, what does the numbers (e.g.: 0.65) next to the line name means?**
>
> **A5.** The numbers displayed next to the line names in Figure 3 correspond to the AUC scores of classification. To ensure clarity, we have included explanatory notes in Figure 3.
>
> **Q6. Based on the models structure (Figure 2) and the description, it is unclear to me which part is outputting the 'predicted labels' in Figure 4.**
>
> **A6.** Thanks for pointing out this issue. The 'predicted labels' are outputted by the Intra-Attribute Classification Module in Figure 2, to assess the model's discrimination of pathology. We will revise this description to make this concept clear.
>
> **Q7. The loss coefficients are set to 0.3, 0.001 and 0.2. Why is it so? Why as low as 0.001?**
>
> **A7.** Thanks for raising this question. (1) The choice of loss coefficients is informed by a meticulous tuning process. To transparently elucidate our reasoning, we provide the parameter sensitivity experiments. For the given optimal coefficients ($\lambda_1$=0.3,$\lambda_2$=0.001,$\lambda_3$=0.2), we first fix the value of $\lambda_2$ and $\lambda_3$, and tune $\lambda_1$ in [0.02,0.03,0.2,0.3,0.5]. Similarly, $\lambda_3$ undergo tuning within the range ([0.02, 0.03, 0.2, 0.3, 0.5]) while keeping optimal $\lambda_1$ and $\lambda_2$ constants. Given the larger magnitude of the ICA loss compared to other losses, $\lambda_2$ was tuned across values ([0.0001, 0.001, 0.01, 0.1]) with the other parameters fixed. The results are listed below, and the metric curves will be added to our final paper.
>
> | $\lambda_1$ | 0.02 |0.03 |0.2 |0.3 |0.5 |
> |--------|--------|--------|--------|--------|--------|
> | Bleu-4 |14.1 | 14.0 |15.1 |**15.5** |14.2 |
> | CIDEr | 18.2 | 19.0 |19.8 |**19.9** |17.4 |
>
> | $\lambda_2$ | 0.0001 |0.001 |0.01 |0.1 |
> |--------|--------|--------|--------|--------|
> | Bleu-4 |14.2 | **15.5** |13.8 |12.9|
> | CIDEr | 17.8 | **19.9** |18.5 |18.0|
>
> | $\lambda_3$ | 0.02 |0.03 |0.2 |0.3 |0.5 |
> |--------|--------|--------|--------|--------|--------|
> | Bleu-4 |14.5 | 14.7 |**15.5** |14.6 |14.2 |
> | CIDEr | 18.2 | 18.5 |**19.9** |18.9 |18.6 |
>
> We employed the most representative Bleu-4 and CIDEr metrics for performance evaluation. We can observe that too low or too high values of $\lambda_1$, $\lambda_2$, $\lambda_3$ decrease the performance. The reason may be because the too low score of $\lambda_1$,$\lambda_2$, and $\lambda_3$ respectively lead to inadequate pathological semantic learning, inter-attribute matching, and visual-textual alignment, which degrades the feature representation. Too high $\lambda_1$,$\lambda_2$, and $\lambda_3$ may weaken the training of report generation-related parameters and cause sub-optimal results.
> (2) As mentioned above, a low value of $\lambda_2$ is used to balance the varying orders of magnitude of model losses. This strategic adjustment ensures that our training procedure converges effectively and yields optimal outcomes, striking a delicate equilibrium.
>
>
> **Q8. Table 1, why there is a line set between up-down and MRMA? Are they belonging to two different groups of models?**
>
> **A8.** We appreciate your inquiry. You've rightly observed that "Up-Down" and "MRMA" originally pertained to distinct tasks. In light of the current limited presence of studies specifically targeting Brain CT report generation - exemplified by the single instance of WGAM - we extended our analysis to encompass a wider scope. We included an array of models from closely related fields such as Chest report generation and image captioning to facilitate a comprehensive assessment, and we separated different groups of models with lines in Table 1.

---

### Official Review · Reviewer_gRk7 · 2023-07-25

**Soundness:** 4

**Excitement:**

4: Strong: This paper deepens the understanding of some phenomenon or lowers the barriers to an existing research direction.

**Missing References:**

N/A

**Paper Topic And Main Contributions:**

This paper proposes a Pathological Graph-driven Cross-modal Alignment (PCGA) model, for Brain CT report generation. In particular, a graph model is used to learn fine-grained visual cues, and algin the cues with text. The structure of the graph model comprises a tissue subgraph and a lesion subgraph, with nodes within and between each subgraph connected according to prior domain knowledge. The PCGA is then further trained with contrastive learning, and compared against multiple general and medical image captioning/report models, on the BCT-CHR brain CT scan dataset.


**Questions For The Authors:**

A. In Section 3.2.3, it is stated that the IAC module dynamically updates node features, for both the tissue nodes, and the lesion nodes. It is also stated that learning inter-attribute relations between tissues and lesions is also essential. This seems to imply that the weights within the tissue subgraph and lesion subgraphs, are also updated/learnt. However, Section 1 stats that the tissue and lesion (sub)graphs are fixed, in contrast to the dynamic tissue-lesion graph. The fixed nature of the tissue and lesion (sub)graphs might thus be clarified - does it mean that the connections are preset, i.e. no new connections can be learnt?

B. Related to the above, it might be clarified as to whether new connections can be learnt for inter-graph weights (i.e. new non-zero values in the sparse attribute alignment matrix)

C. In Section 3.3, a single L_IAC loss is given in Equation 8, while Equation 4 implies two losses (one for each subgraph). This might be clarified. Moreover, if there are indeed two IAC losses, are their weights constrained to be identical?

D. In Section 4.1, it is stated that the reports were in Chinese. It might be clarified as to whether the ensuing training of the model involved the Chinese words/tokens, or English tokens after translation.

E. In Section 4.2, it is stated that 24 keywords were selected to evaluate the CE metrics. It might be discussed as to how these keywords were chosen.

F. In Section 4.3, the data used to tune the loss coefficients and other parameters might be explicitly stated - was it the validation set?

G. In Section 4.4.1, overrepresentation of some high-frequency terms is stated to be a factor possibly affecting CIDEr. It might be briefly discussed as to whether sampling regimes had been considered, to target such token distribution issues.

H. In Section 4.4.2, for the ablation study, it is not immediately clear as to whether any of the configurations represent the proposed model with the initial graph (as designed from prior domain knowledge), but before learning. If the tissue-lesion graph has initial values, it seems that this "original graph" might be evaluated too, and the performance reported. This would establish the contribution of the graph training/updating process.

I. For the CE AUROCs presented in Figure 3, it might be strongly considered to also compare against standard image-based deep learning classification models (e.g. ResNet as already used in this study, etc.) for direct label classification performance.

J. In Section 4.4.3, it is stated that salient regions grounded by the IAC module(s) are generally consistent with human-labeled boxes. The availability of the human annotations appears to admit quantitative analysis, as to performance.


**Reasons To Accept:**

 - Demonstrates potential of incorporating domain knowledge as a specialized graph, for medical report generation from images
 - Extensive comparison against many state-of-the-art models


**Reasons To Reject:**

 - Contribution of further Pathological Graph updating not very clear
 - Minimal comparisons for element-based classification


**Reproducibility:**

4: Could mostly reproduce the results, but there may be some variation because of sample variance or minor variations in their interpretation of the protocol or method.

**Reviewer Confidence:**

4: Quite sure. I tried to check the important points carefully. It's unlikely, though conceivable, that I missed something that should affect my ratings.

**Typos Grammar Style And Presentation Improvements:**

In Line 146, "chesty terminologies" might be rephrased.
In Line 289, "with human refines" might be rephrased.

---

> ### Author Rebuttal · Authors · 2023-08-28
>
> **Q1. Is there a contradiction between Section 3.2.3 and Section 1 regarding the weights within the tissue and lesion subgraphs are dynamically updated or fixed? The fixed nature of the tissue and lesion (sub)graphs might thus be clarified - does it mean that the connections are preset?**
>
> **A1.** Thanks for your constructive comments. The edges (weights) within the tissue and lesion subgraphs are indeed predetermined based on expert medical knowledge, rendering them fixed and unalterable during training. This fixed nature ensures the injection of precise pathological relationships into the graph convolution network, leading to dynamic updates of node features.
> To clarify, the "dynamic update" mentioned in Section 3.2.3 refers only to the tuning of graph node features and does not involve any new connections within the tissue and lesion subgraphs. We will include these analyses to enhance clarity in the final version.
>
>
> **Q2. Related to the above, it might be clarified as to whether new connections can be learnt for inter-graph weights.**
>
> **A2.** Thank you for your valuable feedback. Indeed, new connections can be learned in the tissue-lesion graph. This is achieved by automatically extracting tissue-lesion relationships from the report in each sample, which contributes to generating a tissue-lesion alignment matrix to store the value of edges for each sample. This matrix plays a crucial role in regulating the learning of node features, with non-zero values precisely denoting the existence of tissue-associated lesions in the report.
>
> **Q3. In Section 3.3, a single $L_{IAC}$ loss is given in Equation 8, while Equation 4 implies two losses (one for each subgraph). This might be clarified. Moreover, if there are indeed two IAC losses, are their weights constrained to be identical?**
>
> **A3.** Thanks for your valuable observation. The $L_{IAC}$ loss indeed encompasses two distinct components: one related to tissue attribute and another to lesion attribute. These two attributes hold equal significance in the context of report generation, and as such, we've amalgamated their losses with equal weights in a 1:1 ratio to construct the ultimate $L_{IAC}$ loss. The tuning of the $L_{IAC}$ loss's importance is governed by the parameter $\lambda_1$, as denoted in Equation 8. We will clarify these concerns after Equation 4 in our final version.
>
> **Q4. In Section 4.1, it is stated that the reports were in Chinese. It might be clarified as to whether the ensuing training of the model involved the Chinese words/tokens, or English tokens after translation.**
>
> **A4.** Thanks for pointing these out. The reports in the BCT-CHR dataset are in Chinese, so our model training uses Chinese tokens. English translations in the paper are only for readers' understanding and are not used in training. We will clarify this in the final version.
>
>
> **Q5. In Section 4.2, it is stated that 24 keywords were selected to evaluate the CE metrics. It might be discussed as to how these keywords were chosen.**
>
> **A5.** Thanks for your constructive suggestion. The keywords to evaluate the CE metrics are first filtered as high-frequency tokens in the training corpus and further chosen by experienced radiologists, which represent the most common but critical cranial words and hold great significance within the context of cranial pathology assessment.
>
> **Q6. In Section 4.3, the data used to tune the loss coefficients and other parameters might be explicitly stated - was it the validation set?**
>
> **A6.** Thanks for pointing out this issue. It is true that the hyperparameters in our model are tuned via the validation set, and we will add this description in section 4.3.
>
>
> **Q7. In Section 4.4.1, overrepresentation of some high-frequency terms is stated to be a factor possibly affecting CIDEr. It might be briefly discussed as to whether sampling regimes had been considered, to target such token distribution issues.**
>
> **A7.** Thanks for your valuable comments. For fair model comparisons, we followed the widely used random sampling regime for training. Our future research will explore introducing curriculum learning in data sampling, and an easy-to-hard training paradigm will be developed to smooth the representation of pathological words, which has the potential to tackle this token distribution issue.
>
>
> **Q8. In Section 4.4.2, for the ablation study, the contribution of the graph training/updating process might be evaluated too.**
>
> **A8.** Thanks for your suggestions. Note that the graph node features are only updated by IAC and ICA modules, and the case without training the tissue-lesion graph by ICA module is given in Table 2(e). As suggested, to further demonstrate the benefits brought by graph training, we have added two new experiments in Table 2: (f) the case when the initial graph (tissue and lesion graphs) and the tissue-lesion graph are given but not trained; (g) the case when only tissue-lesion graph is trained. The results are as follows:
>
> | Methods | tissue | lesion | $L_{IAC}$ | $L_{ICA}$ | $L_{CL}$ | B1 | B2 | B3 | B4 | M | RG | C |
> |--------|--------|--------|--------|--------|--------|--------|--------|--------|--------|--------|--------|--------|
> | Baseline | ✖ | ✖ | ✖ | ✖ | ✖ | 40.7 | 27.5 | 19.2 | 13.8 | 26.4 | 35 3 | 16.3 |
> | (a) | ✖ | ✖ | ✖ | ✖ | ✔ | 41.8 | 28.4 | 19.9 | 14.2 | 27.1 | 35.7 | 18.7 |
> | (b) | ✔ | ✖ | ✔ | ✖ | ✖ | 42.5 | 28.9 | 20.3 | 14.6 | 27.3 | 35.7 | 18.0 |
> | (c) | ✔ | ✖ | ✔ | ✖ | ✔ | 43.5 | 29.6 | 20.7 | 14.9 | 27.8 | 35.9 | 19.7 |
> | (d) | ✔ | ✔ | ✔ | ✖ | ✖ | 43.9 | 29.7 | 20.8 | 14.8 | 28.3 | 36.0 | **21.3** |
> | (e) | ✔ | ✔ | ✔ | ✖ | ✔ | 44.1 | 29.8 | 20.8 | 14.8 | 27.8 | 35.9 | 19.3 |
> | (f) | ✔ | ✔ | ✖ | ✖ | ✔ | 41.5 | 27.8 | 19.3 | 13.6 | 26.3 | 35.2 | 16.7 |
> | (g) | ✔ | ✔ | ✖ | ✔ | ✔ | 42.3 | 28.7 | 20.2 | 14.5 | 27.3 | 35.5 | 18.6 |
> | Ours| ✔ | ✔ | ✔ | ✔ | ✔ | **45.0** | **30.8** | **21.6** | **15.5** | **28.7** | **36.5** | 19.9 |
>
> It can be seen that without the graph training, (f) performs even worse than coarse-grained CLIP loss in (a). In contrast, with the IAC, ICA, and both the two modules for graph training, (e), (g), and Ours respectively show improved results, affirming the significance of graph training.
>
>
> **Q9. For Figure 3, it might be strongly considered to also compare against standard image-based deep learning classification models (e.g. ResNet) for direct label classification performance.**
>
> **A9.** Thanks for your constructive comments. As recommended, to further evaluate the classification performance, we compare our model with the standard ResNet model for direct classification. The AUC scores are listed as follows, which will be illustrated in ROC curves and added to Figure 3.
>
> |  | brainstem | midline structure | fourth ventricle  | sulcus cerebri | temporal lobe | thalamus | low density | edema | widen | increase |
> |--------|--------|--------|--------|--------|--------|--------|--------|--------|--------|--------|
> | ResNet | 0.54 | **0.63** | 0.61 | 0.59 | 0.59 | 0.56 | 0.53 | **0.60** | 0.64 | 0.50 |
> | Ours | **0.65** | 0.60 | **0.64** | **0.63** | **0.62** | **0.57** | **0.55** | 0.55 | **0.69** | **0.56** |
>
> It is noticeable that our model outperforms the classic ResNet model in pathology classification, which not only indicates the effectiveness of our graph learning and cross-modal alignment methods but also hints report generation and classification can boost each other.
>
> **Q10. In Section 4.4.3, it is stated that salient regions grounded by the IAC module(s) are generally consistent with human-labeled boxes. The availability of the human annotations appears to admit quantitative analysis, as to performance.**
>
> **A10.** We appreciate your constructive suggestion and understand the importance of a quantitative evaluation to provide a more comprehensive understanding of the model's performance. However, the BCT-CHR dataset does not yet contain the salient objective data, and manually collecting this data would be an incredibly time-consuming and costly endeavor. Nevertheless, our qualitative observations, as presented in the manuscript, indicate that the salient regions grounded by the IAC module(s) are generally consistent with expert-labeled boxes. While we acknowledge the limitations of the qualitative evaluation, we believe it still provides valuable insights into the model's performance in grounding salient regions. In the future, we plan to collaborate with domain experts to gather more human-labeled boxes and pave the way for more comprehensive model evaluation.
>
> **Q11. In Line 146, "chesty terminologies" might be rephrased. In Line 289, "with human refines" might be rephrased.**
>
> **A11.** Thank you for figuring these out. "chesty terminologies" is revised as "chest medical terminologies", and "with human refines" is revised as "with artificial refining".

---

### Official Review · Reviewer_nJDR · 2023-08-06

**Soundness:** 3

**Ethical Concerns:**

Yes

**Excitement:**

4: Strong: This paper deepens the understanding of some phenomenon or lowers the barriers to an existing research direction.

**Paper Topic And Main Contributions:**

This paper proposes a pathological graph-driven cross-modal alignment model for brain CT report generation. First, it constructs a pathological graph that incorporates detailed medical knowledge, allowing for the injection of this knowledge into dedicated node features through graph embedding and updating, thereby achieving fine-grained visual representations. Second, it aligns the learned node features with the corresponding word embeddings by cross-modal contrastive learning, to further boost report generation. Extensive experiments demonstrate that the model achieves superior performance in generating clinically accurate reports.

**Questions For The Authors:**

1 Is there any result on the sensitivity of the lambda_1, lambda_2, lambda_3, training-validation loss / metric curve for equation (8) since these hyperparameters are vital for the model and training?

2 Is the data available or how does the reader reproduce the results? Is it possible to apply the methodology to more general audience of NLP community? Maybe visual-language, vision grounding, visual captioning related tasks can be tried to validate the effectiveness of proposed method.

**Reasons To Accept:**

1 The paper is well-written with vivid figures and examples. Readers can get the ideas, what the paper is doing, results, etc., easily.

2 The graph assisted multimodal training is interesting. From the report / paragraph of text, the knowledge graph can be automatically constructed. Then, intra-attribute and inter-attribute sub-graphs can be obtained. Intra-attribute classification, inter-correlation alignment can be enforced leveraging the knowledge graphs. As far as I know, this is the first work leveraging knowledge graph to improve the current visual-language multimodal learning.

**Reasons To Reject:**

1 The paper is only validated on one dataset, BCT-CHR, which is a medical report related dataset and cannot be publicly available. This strongly limits the impact and audience of the paper. It would be better if the paper could validate on more datasets and tasks.

**Reproducibility:**

3: Could reproduce the results with some difficulty. The settings of parameters are underspecified or subjectively determined; the training/evaluation data are not widely available.

**Reviewer Confidence:**

2: Willing to defend my evaluation, but it is fairly likely that I missed some details, didn't understand some central points, or can't be sure about the novelty of the work.

---

> ### Author Rebuttal · Authors · 2023-08-28
>
> **Q1. Is there any result on the sensitivity of the $\lambda_1$, $\lambda_2$, $\lambda_3$, training-validation loss / metric curve for equation (8) since these hyperparameters are vital for the model and training?**
>
> **A1.** Thanks for your valuable comments. As recommended, we have conducted a thorough analysis to evaluate how different hyperparameters affect our results. Our approach involved conducting several experiments where we fixed two hyperparameters and varied the third within a specific range.
> For the given optimal parameters ($\lambda_1$=0.3,$\lambda_2$=0.001,$\lambda_3$=0.2), we first fix the value of $\lambda_2$ and $\lambda_3$, and tune $\lambda_1$ in [0.02,0.03,0.2,0.3,0.5]. In the same manner, $\lambda_3$ is tuned in [0.02,0.03,0.2,0.3,0.5] with the fixed optimal $\lambda_1$ and $\lambda_2$. Since ICA loss has a greater order of magnitude compared to other losses, we tune the $\lambda_2$ in [0.0001,0.001,0.01,0.1] with the other parameters fixed. The results are presented as follows.
>
> | $\lambda_1$ | 0.02 |0.03 |0.2 |0.3 |0.5 |
> |--------|--------|--------|--------|--------|--------|
> | Bleu-4 |14.1 | 14.0 |15.1 |**15.5** |14.2 |
> | CIDEr | 18.2 | 19.0 |19.8 |**19.9** |17.4 |
>
> | $\lambda_2$ | 0.0001 |0.001 |0.01 |0.1 |
> |--------|--------|--------|--------|--------|
> | Bleu-4 |14.2 | **15.5** |13.8 |12.9|
> | CIDEr | 17.8 | **19.9** |18.5 |18.0|
>
> | $\lambda_3$ | 0.02 |0.03 |0.2 |0.3 |0.5 |
> |--------|--------|--------|--------|--------|--------|
> | Bleu-4 |14.5 | 14.7 |**15.5** |14.6 |14.2 |
> | CIDEr | 18.2 | 18.5 |**19.9** |18.9 |18.6 |
>
> We employed the most representative Bleu-4 and CIDEr metrics for performance evaluation. It can be observed that too low or too high values of $\lambda_1$,$\lambda_2$,$\lambda_3$ decrease the performance. The reason could be that the inadequate pathological graph learning is caused by too low $\lambda_1$,$\lambda_2$, and $\lambda_3$ respectively, which degrades the feature representation. Too high $\lambda_1$,$\lambda_2$, and $\lambda_3$ may weaken the training of report generation task-related model parameters.
> We will include these sensitivity analysis experiments and discussions in our final version.
>
> **Q2. Is the data available or how does the reader reproduce the results? Is it possible to apply the methodology to more general audience of NLP community?**
>
> **A2.** Thanks for your constructive suggestions. We appreciate your interest in the reproducibility of our results and the potential applicability of our methodology to a broader range of NLP tasks.
>
> (1) Regarding the availability of the dataset, it can be accessed by contacting the authors of paper [1]. For researchers working in related fields, such as Chest X-ray report generation, we recommend constructing a similar graph using our method and applying it to their specific tasks. To further facilitate reproducibility and encourage adoption, we will also open source our code upon publication.
> (2) As for the potential of our method to be applied in other NLP domains, we believe that our core concept of utilizing graph nodes and textual words to enhance feature representations can indeed be beneficial to a wider audience, provided that area-specific graph knowledge is available. To validate this assumption, we have conducted new experiments on the Chest X-ray report generation dataset IU-Xray [2]. For time efficiency, we choose the encoder-decoder model with hierarchical attention as Baseline (consistent with that in our article), and add all the proposed modules as Ours. In the absence of guidance from radiologists, we followed the approach outlined in [3] to select graph nodes, and we allowed the model to automatically learn the edges. The results are presented in the table below:
>
> |  | B1 | B2 | B3 | B4 | M | RG | C |
> |--------|--------|--------|--------|--------|--------|--------|--------|
> | Baseline | 39.5 | 27.3 | 20.0 | 14.6 | 18.4 | 35.6 | 40.5 |
> | Ours | **44.2** | **30.6** | **21.8** | **15.3** | **20.4** | **35.7** | **45.6** |
>
> It is worth noting that the edges in this experiment were constructed without the input of chest experts, which might have introduced inaccuracies in the pathological semantics due to incorrectly connected edges. This could explain the inferior performance compared to mainstream Chest report generation models. Nonetheless, our model outperforms the baseline on the IU-Xray dataset, highlighting the effectiveness of our knowledge injection and feature alignment mechanisms.
>
> In the future, we will explore the extensions in more visual-language tasks and contribute to the NLP field.
>
> [1] Yang S, Ji J, Zhang X, et al. Weakly guided hierarchical encoder-decoder network for brain ct report generation[C]//2021 IEEE International Conference on Bioinformatics and Biomedicine (BIBM). IEEE, 2021: 568-573.
>
> [2] Demner-Fushman D, Kohli M D, Rosenman M B, et al. Preparing a collection of radiology examinations for distribution and retrieval[J]. Journal of the American Medical Informatics Association, 2016, 23(2): 304-310.
>
> [3] Zhang Y, Wang X, Xu Z, et al. When radiology report generation meets knowledge graph[C]//Proceedings of the AAAI Conference on Artificial Intelligence. 2020, 34(07): 12910-12917.
>
> **Q3. Ethical Concerns: Yes**
>
> **A3.** We appreciate the concern regarding ethical considerations, and we want to emphasize that this has been a priority throughout our research. The medical images in the BCT-CHR dataset have undergone a thorough process of de-identification to protect participants' privacy. Additionally, we have obtained the required data permissions, ensuring our work meets ethical standards.

---

### Meta-Review · Area_Chair_MUWr · 2023-09-18

**Recommendation:** 4

**Metareview:**

This paper introduces a novel Pathological Graph-driven Cross-modal Alignment (PCGA) model for the generation of brain CT reports. The authors first construct a pathological graph that encodes detailed medical knowledge, which is then integrated into node features via graph embedding and updating, resulting in fine-grained visual representations. The model aligns these learned node features with corresponding word embeddings through cross-modal contrastive learning to enhance report generation. Extensive experiments on the BCT-CHR dataset show the model's superior performance in generating clinically accurate reports.

Main Contributions:

The authors propose a new framework to capture detailed domain knowledge from a Pathological Graph and align fine-grained visual and textual features of pathology.
The paper introduces the concept of decoupling cranial tissues and lesions via a Pathological Graph in the medical report generation area, a first in the field. This approach can handle the fine-grained alignment between long-text reports and multiple scans.
The authors validate the proposed model comprehensively on the BCT-CHR dataset, demonstrating that their method outperforms previous works in medical report generation.
Reasons for Acceptance:

The paper is well-written, with clear figures and examples that make it easy for readers to understand the ideas, methods, and results.
The use of a graph-assisted multimodal training approach is novel and interesting. The knowledge graph, automatically constructed from the report/paragraph of text, allows for intra-attribute classification and inter-correlation alignment.
The novel pathological graph decouples cranial tissues and lesions, enhancing the model's interpretability while also yielding benefits as demonstrated by the experiments.
The paper demonstrates the potential of incorporating domain knowledge as a specialized graph for medical report generation from images.
The model was extensively compared against several state-of-the-art models, showing superior performance.

---

### Decision · Program_Chairs · 2023-10-07

**Decision:**

Accept-Main

**Comment:**

This paper introduces a novel Pathological Graph-driven Cross-modal Alignment (PCGA) model for the generation of brain CT reports. The authors first construct a pathological graph that encodes detailed medical knowledge, which is then integrated into node features via graph embedding and updating, resulting in fine-grained visual representations. The model aligns these learned node features with corresponding word embeddings through cross-modal contrastive learning to enhance report generation. Extensive experiments on the BCT-CHR dataset show the model's superior performance in generating clinically accurate reports.

Main Contributions:

The authors propose a new framework to capture detailed domain knowledge from a Pathological Graph and align fine-grained visual and textual features of pathology.
The paper introduces the concept of decoupling cranial tissues and lesions via a Pathological Graph in the medical report generation area, a first in the field. This approach can handle the fine-grained alignment between long-text reports and multiple scans.
The authors validate the proposed model comprehensively on the BCT-CHR dataset, demonstrating that their method outperforms previous works in medical report generation.
Reasons for Acceptance:

The paper is well-written, with clear figures and examples that make it easy for readers to understand the ideas, methods, and results.
The use of a graph-assisted multimodal training approach is novel and interesting. The knowledge graph, automatically constructed from the report/paragraph of text, allows for intra-attribute classification and inter-correlation alignment.
The novel pathological graph decouples cranial tissues and lesions, enhancing the model's interpretability while also yielding benefits as demonstrated by the experiments.
The paper demonstrates the potential of incorporating domain knowledge as a specialized graph for medical report generation from images.
The model was extensively compared against several state-of-the-art models, showing superior performance.